# Effects of maternal anemia on low-birth-weight in Sub-Sahara African countries: Systematic review and meta-analysis

Nigus Kabtu Belete[1]*, Abebe Gedefaw Belete[1], Darik Temesgen Assefa[1],
Muluken Bekele Sorrie[1,2], Manaye Yihune Teshale[1,3]

1 Department of public health, College of Medicine and Health Science, Arba Minch University, Arbaminch, Ethiopia, 2 Department of Public Health and Primary Care, Faculty of Medicine and Health Sciences, Ghent University, Gent, Belgium, 3 Department of Health Promotion, CAPHRI Care and Public Health Research Institute, Maastricht University, Maastricht, The Netherlands

* nigushabtu2014@gmail.com

## Abstract

### Introduction

Maternal anemia is a major public health concern that affects women globally, with a particularly high prevalence in developing countries, notably in sub-Saharan Africa. This condition is linked to negative birth outcomes, with low birth weight being a common consequence of maternal anemia during pregnancy. Therefore, this study aims to evaluate the effect of maternal anemia on low birth weight in the context of sub-Saharan African countries.

### Methods

This study involved searching electronic databases, including PubMed, Embase, Scopus, Cochrane, and Web of Science, as well as reference lists and citation tracking for additional studies. It included cohort, case control, and cross-sectional studies published in English between January 2015 and June 2024. Data were extracted using Covidence and transferred to Microsoft Excel, then to Stata™ Version 17.0 for analysis. Heterogeneity was assessed with forest plots and the Inverse variance ($I^2$) test. Subgroup analysis, sensitivity analysis, and meta-regression were performed to explore sources of heterogeneity, while funnel plot symmetry was evaluated for publication bias. The meta-analytic effect was summarized using pooled odds ratios, 95% confidence intervals, and $I^2$ tests for heterogeneity. This was registered on the PROSPERO under the identification number CRD42024561098.

### Result

A total of 1213 articles were identified, 71 of which were screened for full-text review, and 21 involving women from sub-Saharan African countries met the inclusion

**Data availability statement:** All relevant data are within the paper and its Supporting information files.

**Funding:** The author(s) received no specific funding for this work.

**Competing interests:** The authors declare no competing interests.

**Abbreviations:** JBI, Joanna Briggs Institute; LBW, Low Birth Weight; OR, Odd Ratio; RR, Relative Risk; SDG, Sustainable Developmental Goal; SSA, Sub-Sahara Africa; WHO, World Health Organization.

criteria, and were included in this meta-analysis. Women with anemia during pregnancy are at higher risk of giving birth to babies with low birth weight compared to women without anemia (AOR = 3.37; 95% CI: 2.66–4.27; $I^2$: 96.71%).

## Conclusion

Maternal anemia during pregnancy was identified as a significant risk factor for low birth weight. Such that the incidence of low birth weight could possibly be reduced with early identification and proper care of anemia during pregnancy.

## Introduvction

Anemia is a significant global public health concern affecting pregnant women in both developed and developing nations, with a worldwide prevalence of approximately 29.9% [1]. Developing countries, particularly those in Sub-Saharan Africa (57.0%), bear the highest burden. This poses a significant public health challenge, contributing to increased maternal mortality and morbidity, and adverse birth outcomes including low birth weight (LBW) [1,2]. Maternal anemia during pregnancy increases the risk of low birth weight, particularly where nutritional deficiencies are common and prenatal care is limited [1,3].

The World Health Organization (WHO) defines LBW as new-born infants weighing less than 2500 grams at birth, regardless of gestational age [4]. This condition often leads to long-term health complication and increases the risk of chronic illnesses in later life, highlighting the need for effective public health initiatives [5]. Finding from different studies indicate that LBW infants more likely to die than those with higher birth weights [6].

Globally, LBW affects more than 20% of newborns, equating to over 20 million births each year [7]. Poor maternal nutrition during pregnancy can significantly impact fetal growth and birth weight [8]. Adequate nutrition for mothers is essential, as insufficient intake is a major cause of LBW [9]. In 2012, the World Health Assembly established a goal to reduce LBW prevalence by 30% until 2025 [10]. The Sustainable Development Goals (SDGs) aim to reduce the neonatal mortality rate to below 1.2% by 2030, by ensuring healthy lives and well-being for all ages. Reducing LBW is vital to achieving this target, as it significantly contributes to neonatal mortality [11].

Due to a lack of recent review studies particularly after the endorsement of SDG on the link between maternal anemia during pregnancy and LBW through cross sectional, case control and cohort studies in SSA countries. This article aims to investigate the relationship between LBW and maternal anemia systematically evaluate this relationship as studied in multiple countries in SSA. Clarification will be made by a systematic review of the evidence base of journals and abstracts in this topic area, looking at all designs of study.

## Objective

• To assess the effects of maternal anemia on low-birth-weight in Sub-Sahara African countries, 2024.

## Methods and materials

### Protocol

The review protocol was registered on the PROSPERO international database for systematic reviews and meta-analysis under the identification number CRD42024561098.

### Information source and search strategy

We carried out a systematic review and meta-analysis of eligible published and unpublished studies to determine the effect maternal anemia on low-birth-weight SSA countries. The Preferred Reporting Items for Systematic Reviews and Meta-Analyses (PRISMA) [12] guideline is followed for the scientific rigor of the study. The studies for this review were retrieved through reproducible and comprehensive electronic searching of major reputable databases (PubMed, EMBASE, Cochrane library, Scopus and web of science) from June 12/2024 to June 30/2024.

   Afterward, hand-searching of reference lists and citation tracking of included studies was screened for additional studies. The search was performed independently by two authors (NK and MY) using three main categories of keywords related to the review question, Medical Subject Headings (MeSH), and employing Boolean search operators "OR", and "AND". The main keywords utilized for the article search included: maternal anemia, low birth weight, and Sub-Saharan African countries. These keywords were selected because they aligned with the core concepts of the review question. Various combinations and variations of these keywords were employed. We explored synonyms for each search term in the literature to create a comprehensive search strategy. The full search strategy employed and search results identified across the 5 databases are available as a supplementary file (S1 Table).

### Inclusion and exclusion criteria

We included articles that met the following criteria by using PICO format

• Population, or participants- Infant with the diagnosis of LBW;

• Interventions or exposures-Pregnant mother with anemia

• Comparisons or control groups-Pregnant mother without anemia

• Outcomes of interest-LBW < 2500gram, Hemoglobin level <11 g/dl

• Setting- SSA countries.

• Study designs- Retrospective or prospective Cohort, case control and cross-sectional study design.

   We excluded articles with the following criteria

• Studies that did not provide data on the effect of maternal anemia on LBW;

• Ambiguous study designs and outcome measures of no interest; and

• Abstracts, case reports, systematic review studies, case series studies, editorials, review articles, or non-English language articles.

• Study period before January 2015.

## Data extraction

The data were extracted independently by three authors (AG, NG and DT) from eligible studies using a customized data extraction form. The form was filled with information about the author's name, year of publication, place and year of study, objective, study design, sample size, data collection location, data source, criteria for anemia diagnosis, frequency of maternal anemia, percentage of low-birth-weight infants, association measurements and confidence intervals, and confounding variables. If further information or clarification was needed, the primary author of the original article was contacted through email. The article was excluded if, after at least two email attempts, the author did not respond.

The search results were transferred to Covidence, which removed duplicates and evaluated the remaining in two distinct screening steps. The first step involved reviewing the title and abstract to filter relevant records and exclude irrelevant ones based on the review question and eligibility criteria. Three reviewers (AG, NK, and DT) were involved in this step, with each article independently assessed by two reviewers. In case of disagreements, we thoroughly examined the issues and consulted a third-party (MY) to resolve them. The same reviewers conducted the second step based on the full texts. All studies identified in the literature search is available in the (S1 File). All of the data were extracted by three reviewers. Any studies that the reviewers agreed upon were included, and any disagreements were resolved through discussion and consensus to minimize bias (S2 Table).

## Data quality

Two independent reviewers (NK and DT) were assessed the quality of included studies with any disagreements between authors resolved through discussion. We used a specific checklist to assess the methodologic quality of all included cohort, case control and cross-sectional studies with the use of Joanna Briggs Institute (JBI) checklist [13] (S2 File). The methodological quality of the studies was considered poor (high risk of bias if 20–50% items scored yes, moderate risk of bias if 50–80% items scored yes, and Good (low risk of bias if 80–100% items scored yes as per JBI checklist) (S3 Table) [14].

## Data synthesis and items

Hemoglobin levels were categorized as normal if above 11 g/dL and as maternal anemia if below 11 g/dL. LBW) was defined as an infant weight of less than 2500 grams. In this study, the articles were screened by title and abstract using Covidence screening software. Selected records were undergone full text review using the same software. Extracted data was included study type, author name, publication year, number of participants, and pregnancy outcome (LBW). STATA Version 17 was used for data analysis and $p < 0.05$ will be considered significant for all analyses.

The minimum and maximum relative risks (RR) or odds ratio (OR) at upper and lower limits of 95% confidence intervals was extracted for each study. In studies, the adjusted RR was converted in to AOR by using the following formula.

$RR = OR/(1 - p0 + (p0 \times OR))$, Where $p^0$ is the base-rate risk – the probability of the event without the intervention [15]. Meta-regression examined the relationship between study year, effect size, and sources of heterogeneity. Publication bias was assessed using funnel plots. Sensitivity analysis evaluated the influence of individual studies on the overall results. Heterogeneity was assessed using I² and p-values, and a random effects model was applied when heterogeneity was present. Subgroup and sensitivity analyses identified influential risk factors, considering evidence of publication bias.

## Results

### General characteristics and quality of studies

Five databases were searched from June 12 to June 30, 2024, giving 51,923 study participants. After removing 922 duplicates using Covidence, 1,213 records remained. Title and abstract screening, conducted in duplicate, narrowed this down

to 72 records for full-text screening. Of these, 51 articles were excluded based on eligibility criteria, resulting in 21 articles included in this systematic review (Fig 1). Full data extracted from each study for the reported systematic review and/or meta-analysis is available in the (S3 File).

Of the total number of selected studies, 8 were case control, 1was retrospective cohort study and 12 were cross sectional studies. Much of the research was done by cross sectional study design. Three of the studies were carried out in west Africa countries (1 in Gahanna, 1 in Sierra Leone, and 1 in Nigeria) and others were from east Africa (1 in the Sudan, 1 in the Tanzania, 1 in Rwanda and 15 in Ethiopia). The quality of 20 studies was good and only one was moderate according to JBI checklist (Table 1).

## Maternal anemia and low birth weight

By using a random effects model, this study showed that women with anemia during pregnancy are at higher risk of giving birth to babies with low birth weight compared to women without anemia (OR= 3.37; 95% CI: 2.66–4.27). It also showed that the Tadesse et al. (2023) had the highest weight among these studies. As shown in the following (Fig 2), Chi-square

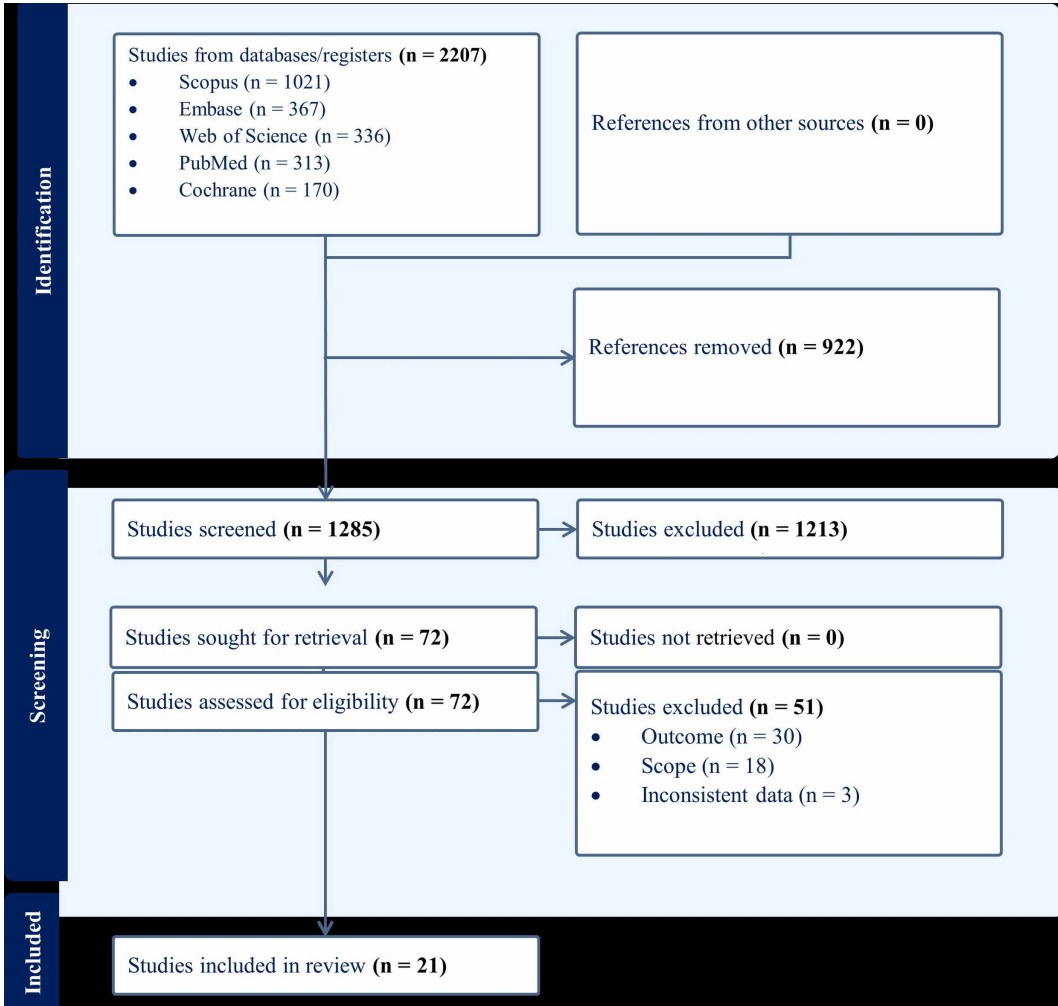

**Fig 1. Flow diagram of article search and screening process for effect of maternal anemia on LBW in SSA countries.**

**Table 1. Study characteristics of eligible studies on effect of maternal anemia on low birth weight in Sub-Sahara countries.**

| Author year | Countries | Study Design | Sample size | Number of Anemic mothers | Number of LBW in anemic mothers | Quality of study |
|---|---|---|---|---|---|---|
| Engidaw et al. (2022) | Ethiopia | Cross sectional | 211 | 34 | 10 | Good |
| Seid et al. (2022) | Ethiopia | Case control | 252 | 39 | 28 | Good |
| Elmugabil et al. (2023) | Sudan | Cross sectional | 253 | 132 | 32 | Good |
| Ahmed et al. (2018) | Ethiopia | Case control | 279 | 47 | 30 | Good |
| Girma et al. (2019) | Ethiopia | Case control | 279 | 47 | 30 | Good |
| Mekie and Taklual (2019) | Ethiopia | Cross sectional | 282 | 11 | 7 | Good |
| Gebrehawerya et al. (2018) | Ethiopia | Case control | 287 | 33 | 26 | Good |
| Mingude et al. (2020) | Ethiopia | Case control | 300 | 56 | 27 | Good |
| Lake and Fite (2019) | Ethiopia | Cross sectional | 304 | 108 | 26 | Good |
| Aboye et al. (2018) | Ethiopia | Cross sectional | 308 | 29 | 12 | Good |
| Tadesse et al. (2023) | Ethiopia | Cross sectional | 337 | 57 | 18 | Good |
| Abera et al. (2019) | Ethiopia | Cross sectional | 358 | 66 | 29 | Good |
| Adam et al. (2019) | Ghana | Case control | 360 | 193 | 78 | Good |
| Hailu et al. (2021) | Ethiopia | Cross sectional | 363 | 43 | 16 | Good |
| Kumlachew et al. (2018) | Ethiopia | Cross sectional | 375 | 65 | 28 | Good |
| Muluneh et al. (2023) | Ethiopia | Cross sectional | 422 | 81 | Not provided | Good |
| Kargbo et al. (2021) | Sierra Leone | Case control | 438 | 146 | 82 | Good |
| Deriba and jemal (2021) | Ethiopia | Case control | 555 | 104 | 64 | Good |
| Oladeinde et al 2015 | Nigeria | Cross sectional | 780 | 264 | 28 | Good |
| Biracyaza et al. (2021) | Rwanda | Cross sectional | 7381 | 1982 | 257 | Good |
| Mitao et al. (2016) | Tanzania | Retrospective Cohort | 37799 | 686 | 122 | Moderate |

tests show that the results of the studies were heterogeneous and significant ($I^2$ Index = 96.7% P = 0.000). When there is heterogeneity among results of studies that means the results of studies are different.

## Publication bias

The publication bias was evaluated with a funnel plot, which showed that the study on the relationship between maternal anemia and, LBW, has a publication bias which is indicated by the asymmetry of the study by looking at the number of dots on the right and left sides and comparing it with the standard error (Fig 3).

## Sub group analysis

Subgroup analysis on the relationship has been evaluated by 7 sub-Sahara Africa countries study. By combining Ethiopian studies using a random effects model, maternal anemia was significantly associated with LBW (AOR, 4.34 [95% CI: 4.24 to 4.43]). This showed that the study conducted in Ethiopia have been significancy relationship (Fig 4).

Another subgroup analysis was performed to determine the effect of maternal anemia on LBW in SSA based on the study year of publication. Studies published from 2015 to 2019 had the greatest significant at 2.91 AOR (95%CI: 2.03–4.17) with heterogeneity $I^2$ = 95.54%. However, studies published from 2020–2024 had less heterogeneity with random effect model (Fig 5).

We also performed subgroup analysis based on study design showed that study done by using cross sectional study design indicated strong association between maternal anemia and LBW (AOR:3.64, 95%CI: (2.65–5.08)) with heterogeneity of $I^2$ = 96.21% (Fig 6).

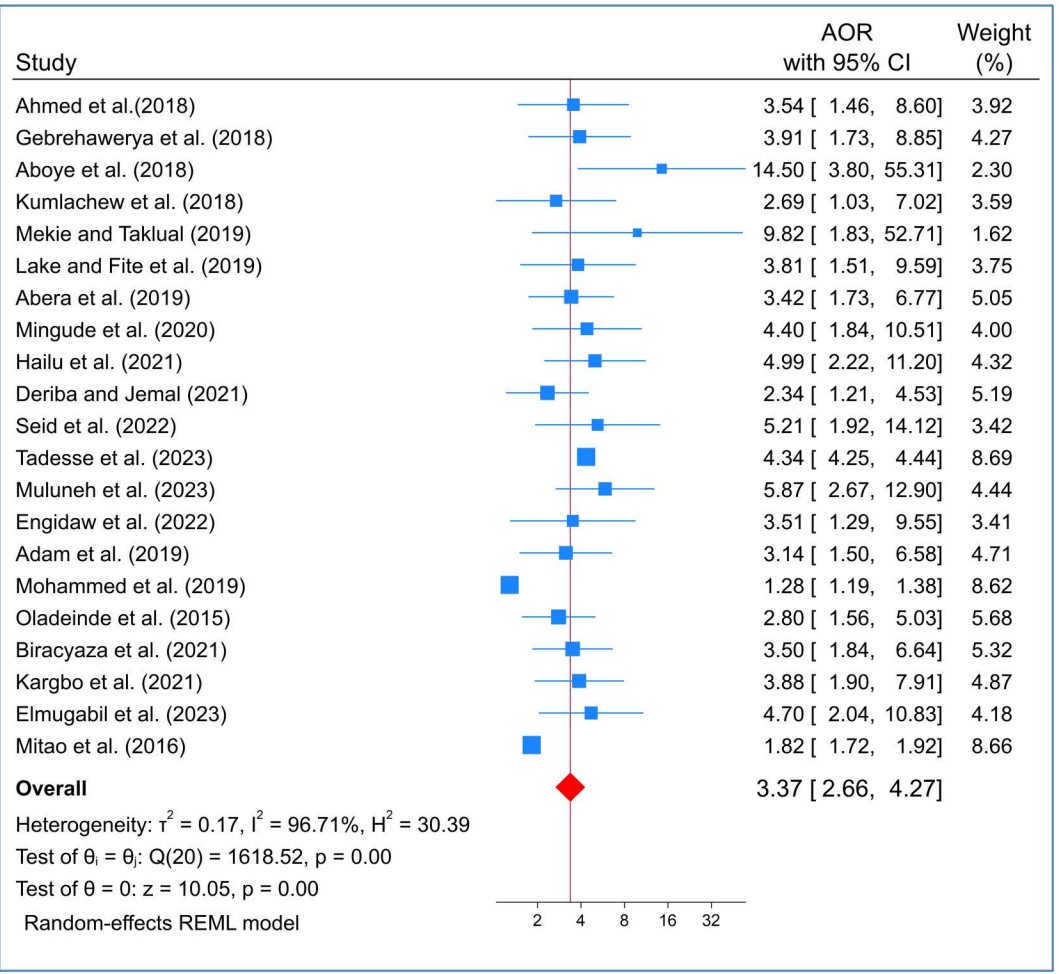

**Fig 2. Forest plot of the pooled effects of maternal anemia on LBW in SSA countries.**

## Sensitivity analysis

Sensitivity analysis revealed that removing a study can either affect the results or not. It highlights the influence of individual studies in meta-analysis, as the removal of a study can significantly alter the overall effect. In this study as shown in the following figure (Fig 7), eliminating a study had no effect on the overall results.

## Discussion

This systematic review and meta-analysis showed the effect of maternal anemia during pregnancy on LBW. Maternal anemia is a major public health concern globally, specifically in developing countries and sub-Saharan Africa (SSA), where the pooled prevalence among pregnant women is 51.26%. This problem leads to adverse birth outcome like LBW [16].

This systematic review and meta-analysis reviled that infant who born from anemic mother had three times (AOR:3.37 (95% CI: 2.66, 4.27) more likely to have LBW compared with those whose mothers did not have anemia during their pregnancy. There is still much to learn about the biological plausibility of the association between low/insufficient birth weight and maternal anemia [17]. However, other studies have indicated that anemia in the mother may have an impact on the birth weight by increasing the risk of intrauterine growth restriction for the

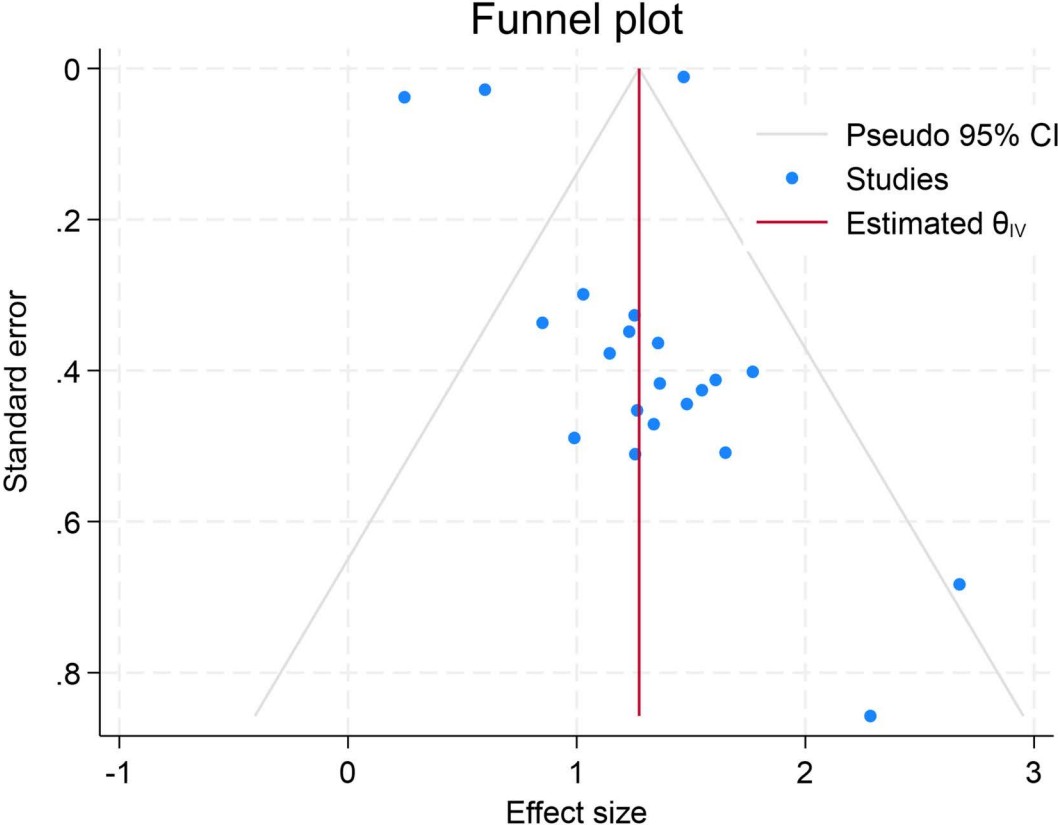

**Fig 3. Funnel plot of the pooled effects of maternal anemia on LBW in SSA countries.**

fetus (IUGR) [18]. During the mid-second trimester, pregnant women typically experience hypervolemia with a 30–40 ml/kg increase in plasma volume. If red blood cell production does not match this expansion, hemodilution and maternal anemia may result [19]. This causes the usual plasma volume and red cell mass to grow [20]. These changes have an effect on the variation in hemoglobin levels. The fetal birth weight may then be impacted by the mother's hemoglobin content dropping. The finding of this study is supported by other systematic review and meta-analysis research findings which showed that anemia during pregnancy was significantly associated with LBW (RR = 1.31,95% CI, (1.13–1.51)) [21], (RR = 1.28, 95% CI (1.10–1.50)) [22], (OR=1.23 95%, CI (1.06–1.43)) [23], (OR=3.42, 95% CI (1.85–6.34)) [3], and (OR=1.9, 95%, CI (1.06–2.60)) [24]. However, the degree of association was different for each study. The discrepancy for this degree of association may be difference in study context. Most of the study was done in global population [3,22,23], one was done in South Asian countries [24], and one was done in low- and middle-income countries [21]. Another explanation for the deference might be due to difference in outcome measures. In this study the outcome measures were adjusted odd ratio but other study used relative risk [21,22], or crude odd ratio [3,23,24].

The analysis of sub-groups was shown that significant relationships have been observed more in studies done in Ethiopia among pregnant women with anemia (AOR: 4.34, 95% CI: (4.24–4.43)). Most of the study for this review came from Ethiopia; LBW occurrences were widespread in the research, with a noticeable association noted particularly in developing regions such as Asia and Africa [21,22]. However, no study conducted a subgroup analysis based on countries. Another subgroup analysis conducted based on publication year indicated that articles published

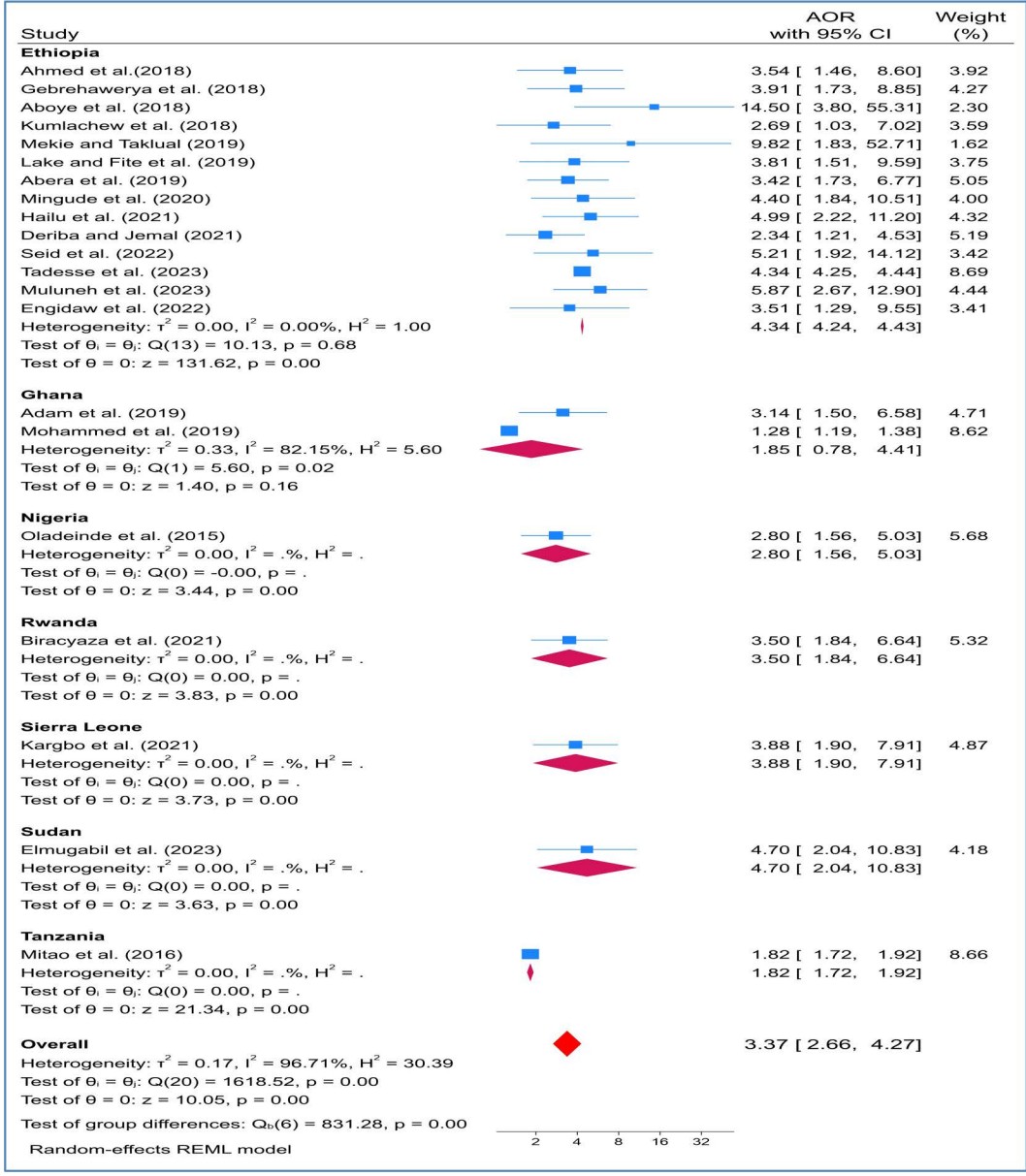

**Fig 4. Subgroup analysis by countries on effects of maternal anemia on LBW in SSA countries.**

between 2015–2019 demonstrated a stronger association between maternal anemia and LBW (AOR: 2.91, 95% CI 2.03–4.71). This timeframe marks the early implementation of SDG and other initiatives aimed at reducing maternal and infant mortality and morbidity. It is expected that the prevalence of this issue will decrease over time with various intervention strategies. Similarly, research utilizing a cross-sectional study design showed a significant relationship with LBW (AOR: 3.64, 95% CI 2.62–5.08). These results contrast with a study conducted in South Asian nations, which found no substantial variations in LBW based on study designs [24]. This discrepancy could be attributed to differences in populations, as the former study focused on South Asian countries while our research pertains to SSA countries. Additionally, it may be influenced by the types of studies included, as most articles considered all available

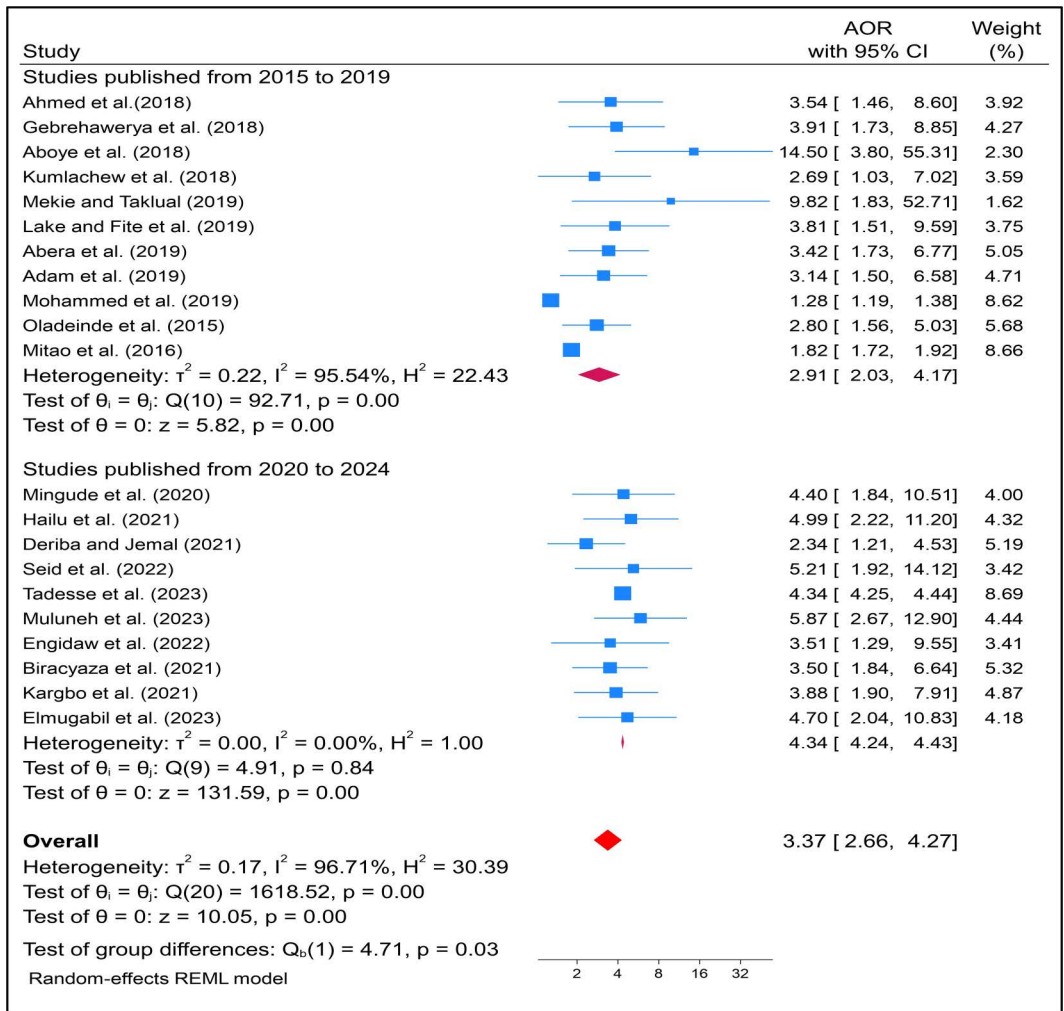

**Fig 5. Subgroup analysis by publication year on effects of maternal anaemia on infant low birth weight in SSA countries.**

research whereas our study specifically analyzed studies conducted between 2015 and 2024, post the inception of the SDG program.

## Conclusion

This review implies that anemia is a critical public health burden among pregnant mothers that can affect the weight of newborns. The odd of LBW was nearly three times more common in anemic mothers. These suggest the strengthen of anemia screening during pregnancy for early diagnosis and management that may deter the progression of LBW and related complications.

## Limitations of the study

There are a few notable limitations to this systematic review and meta-analysis. Firstly, there is a lack of consistency in anemia cut-off points across the studies, leading to the exclusion of many studies. Secondly, applying the conclusions from a limited number of sub-Saharan African nations included in this analysis to the entire region may be challenging.

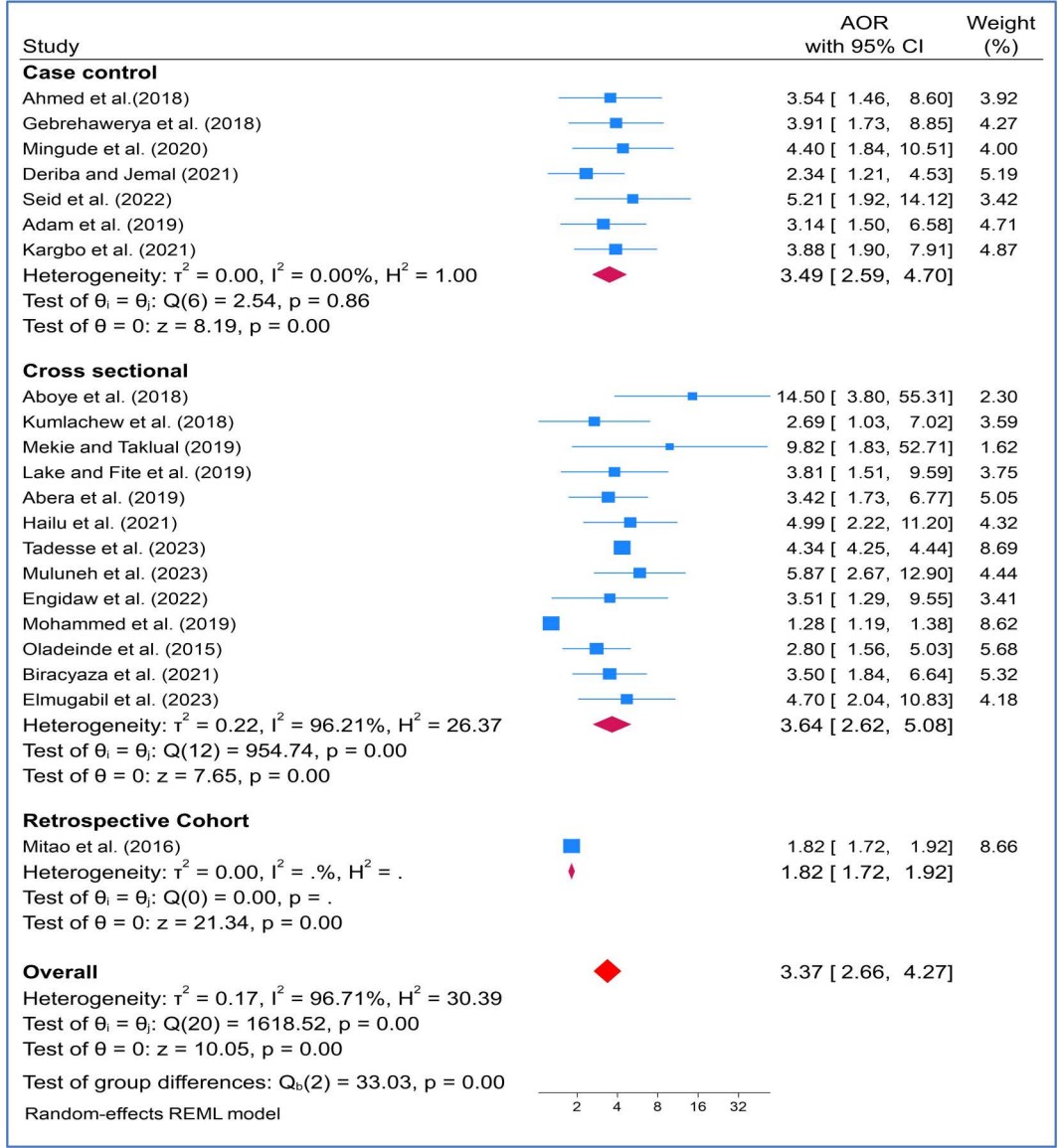

**Fig 6. Subgroup analysis by study design on effects of maternal anaemia on infant low birth weight in SSA countries.**

Thirdly, most of the reviewed studies were cross-sectional, limiting the ability to establish a cause-effect relationship. Therefore, further studies with robust designs considering various causes of anemia are warranted. Moreover, significant heterogeneity among the included studies and the underlying causes for this heterogeneity remain unclear, potentially complicating result interpretation.

## Supporting information

**S1 Table. The search strategy and search results of identified databases.**
(DOCX)

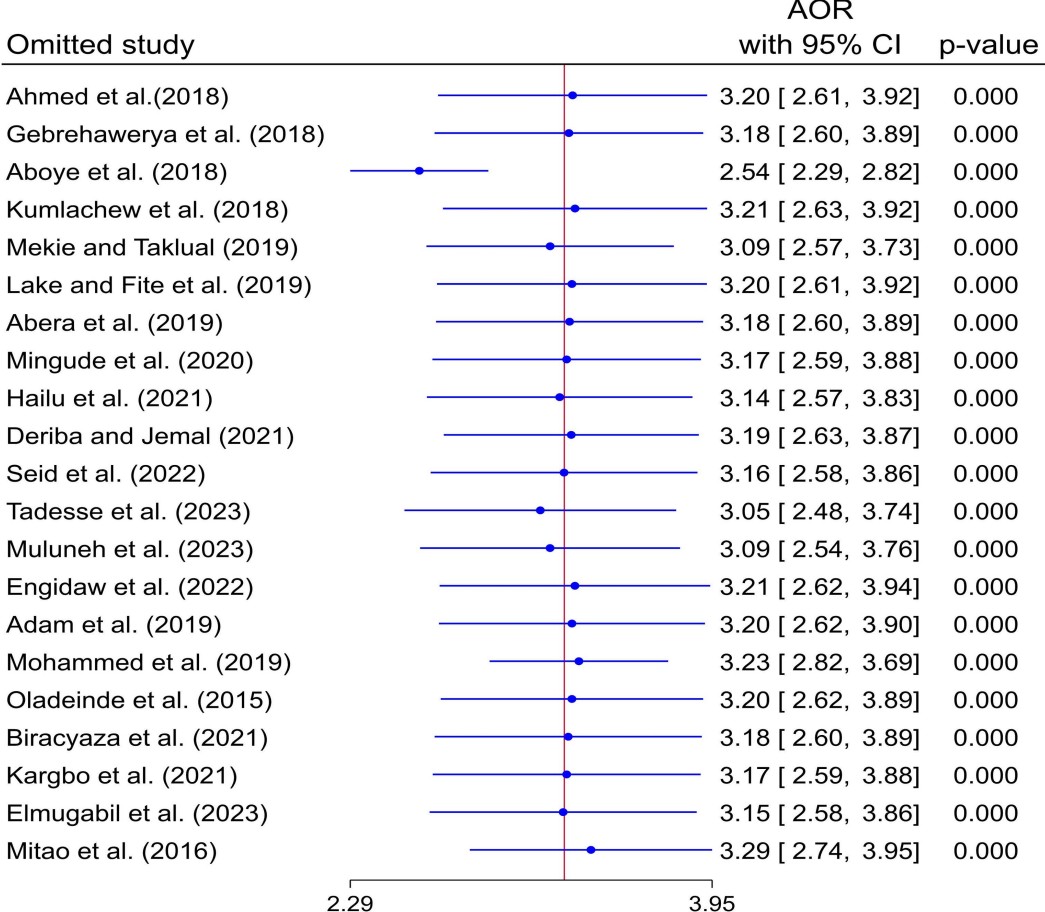

**Fig 7. Sensitivity analysis on the effect of maternal anemia on LBW in SSA countries.**

**S2 Table. All data extracted from each study for the reported systematic review and meta-analysis.**
(DOCX)

**S3 Table. Risk of bias and quality/certainty assessments for included studies.**
(DOCX)

**S1 File. All studies identified in the literature search, including those that were excluded from the analyses with reason of exclusion.**
(XLSX)

**S2 File. Joanna Briggs Institute (JBI) checklist to assess the methodologic quality of all studies.**
(PDF)

**S3 File: Full datasets analyzed during the current study.**
(XLSX)

## Author contributions

**Conceptualization:** Nigus kabtu Belete, Darik Temesgen Assefa, Manaye Yihune Teshale.

**Data curation:** Abebe Gedefaw Belete, Darik Temesgen Assefa, Muluken Bekele Sorrie, Manaye Yihune Teshale.

**Formal analysis:** Nigus kabtu Belete, Abebe Gedefaw Belete, Darik Temesgen Assefa, Muluken Bekele Sorrie, Manaye Yihune Teshale.

**Investigation:** Abebe Gedefaw Belete.

**Methodology:** Nigus kabtu Belete, Muluken Bekele Sorrie, Manaye Yihune Teshale.

**Resources:** Darik Temesgen Assefa.

**Software:** Manaye Yihune Teshale.

**Supervision:** Manaye Yihune Teshale.

**Writing – original draft:** Nigus kabtu Belete, Manaye Yihune Teshale.

**Writing – review & editing:** Nigus kabtu Belete.

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
