## [Decision Letter · Decision Letter 0]

PONE-D-24-38352Effects of Maternal Anemia on low-birth-weight in Sub-Sahara African countries: systematic review and meta-analysisPLOS ONE

Dear Dr. Belete,

Thank you for submitting your manuscript to PLOS ONE. After careful consideration, we feel that it has merit but does not fully meet PLOS ONE’s publication criteria as it currently stands. Therefore, we invite you to submit a revised version of the manuscript that addresses the points raised during the review process.

Please note that we have only been able to secure a single reviewer to assess your manuscript. We are issuing a decision on your manuscript at this point to prevent further delays in the evaluation of your manuscript. Please be aware that the editor who handles your revised manuscript might find it necessary to invite additional reviewers to assess this work once the revised manuscript is submitted. However, we will aim to proceed on the basis of this single review if possible.

We look forward to receiving your revised manuscript.

Kind regards,

Emma Campbell, Ph.D

Staff Editor

PLOS ONE

Journal Requirements:

-Prognostic role of vitamin D receptor in breast cancer: a systematic review and meta-analysis (DOI: 10.1186/s12885-020-07559-w)

Perinatal outcomes in anemic pregnant women in public hospitals of eastern Ethiopia (https://doi.org/10.1093/inthealth/ihac021)

(Among others)

In your revision ensure you cite all your sources (including your own works), and quote or rephrase any duplicated text outside the methods section. Further consideration is dependent on these concerns being addressed.

3. As required by our policy on Data Availability, please ensure your manuscript or supplementary information includes the following:

5. In the online submission form, you indicated that the datasets analyzed during the current study are available from the corresponding author on reasonable request.

Reviewers' comments:

Reviewer's Responses to Questions

**Comments to the Author**

1. Is the manuscript technically sound, and do the data support the conclusions?

Reviewer #1: Partly

2. Has the statistical analysis been performed appropriately and rigorously? 

Reviewer #1: Yes

3. Have the authors made all data underlying the findings in their manuscript fully available?

Reviewer #1: Yes

4. Is the manuscript presented in an intelligible fashion and written in standard English?

Reviewer #1: Yes

5. Review Comments to the Author

Reviewer #1: question 4 comment: Requires careful editing for proper English grammar

The sizable impact of low birth weight in neonates and its link to mortality and morbidity has long been known.

Many published papers, but certainly not all, have reported associated between iron deficiency anemia in pregnancy and low birth weight. It is difficult for a reader to determine if the meta-analysis and systematic review was focused on anemia or on iron deficiency anemia.

The authors correctly point to the importance of adequate nutrition as a preventive measure to reduce low birth weight.

Rates of iron indices in pregnancy vary greatly among African countries. These rates are impacted by factors such as iron in groundwater, a deficiency in related micronutrients, the presence of helminths, HIV/AIDS, body weight, age, prior pregnancy history, and perhaps most importantly the hemoglobin level (and ferritin) present when a woman enters pregnancy.

Of the 21 studies reviewed, it is important to note that none were randomized, placebo-controlled trials. While the association of anemia to lower birth weight in anemic women during pregnancy appears to be confirmed, it is difficult to extrapolate the importance of these findings on actionable approaches to care.

While the standards for a systematic review and metanalysis are well maintained, the reported findings may not be generalizable.

6. PLOS authors have the option to publish the peer review history of their article (what does this mean? ). If published, this will include your full peer review and any attached files.

**Do you want your identity to be public for this peer review?** For information about this choice, including consent withdrawal, please see our Privacy Policy .

Reviewer #1: No

---

## [Author Response · Author response to Decision Letter 1]

26 Mar 2025

Thank you for your insightful and supportive feedback. We have addressed all concerns raised by the editor and reviewer, and I am happy to clarify any remaining issues in the future.

S.no Editor comments and Authors response

1. Please ensure that your manuscript meets PLOS ONE's style requirements, including those for file naming. Accepted and corrected according to the guideline.

2 We noticed you have some minor occurrence of overlapping text with the following previous publication(s), which needs to be addressed: Accepted and corrected: We tried to paraphrase again.

3 A numbered table of all studies identified in the literature search, including those that were excluded from the analyses.

For every excluded study, the table should list the reason(s) for exclusion. Accepted: We add S 2 Table as a supporting information that include all studies including the exclude one and reason for exclusion.

All data extracted from each study for the reported systematic review and/or meta-analysis that would be needed to replicate your analyses. Accepted: We add S 4 Table as a supporting information have all extracted data, confirmation for eligibility, name of data extractor, and data of data extraction.

4 We note that you have indicated that there are restrictions to data sharing for this study. Corrected: The datasets analyzed during the current study are available in S5 file and attached as a supporting information.

5 In the online submission form, you indicated that the datasets analyzed during the current study are available from the corresponding author on reasonable request.

6 Please include captions for your Supporting Information files at the end of your manuscript, and update any in-text citations to match accordingly. Accepted and corrected: We put the citations of all supporting information at the end of the manuscript below the reference and update the citation.

Reviewer Comments and Author Responses

1. Reviewer #1: question 4 comment: Requires careful editing for proper English grammar Accepted: We tried to edit each sentence and paragraph throughout the document.

2. The sizable impact of low birth weight in neonates and its link to mortality and morbidity has long been known.

Many published papers, but certainly not all, have reported associated between iron deficiency anemia in pregnancy and low birth weight. It is difficult for a reader to determine if the meta-analysis and systematic review was focused on anemia or on iron deficiency anemia. Accepted and corrected: Thank you for your review. A paragraph describing anemia burden and consequences has been added (P3 L48-54). This study focuses on anemia generally and does not assess iron deficiency anemia independently.

3. The authors correctly point to the importance of adequate nutrition as a preventive measure to reduce low birth weight.

Rates of iron indices in pregnancy vary greatly among African countries. These rates are impacted by factors such as iron in groundwater, a deficiency in related micronutrients, the presence of helminths, HIV/AIDS, body weight, age, prior pregnancy history, and perhaps most importantly the hemoglobin level (and ferritin) present when a woman enters pregnancy. I appreciate your concern and agree with your suggestions. The manuscript highlights nutritional intervention packages as a preventive measure, alongside other cause-specific interventions like infection prevention and management.

4. Of the 21 studies reviewed, it is important to note that none were randomized, placebo-controlled trials. While the association of anemia to lower birth weight in anemic women during pregnancy appears to be confirmed, it is difficult to extrapolate the importance of these findings on actionable approaches to care.

While the standards for a systematic review and metanalysis are well maintained, the reported findings may not be generalizable. Thank you for your concern and pertinent questions. This study included cohort, case-control, and cross-sectional designs, but excluded experimental designs. While experimental studies offer high-quality evidence, their interventional nature, where an effect follows a specific action, makes them difficult to integrate with observational study designs. Therefore, the best approach was to exclude them or conduct a separate with only similar experimental studies.

---

## [Decision Letter · Decision Letter 1]

Effects of Maternal Anemia on low-birth-weight in Sub-Sahara African countries: systematic review and meta-analysis

PONE-D-24-38352R1

Dear Dr. Belete,

We’re pleased to inform you that your manuscript has been judged scientifically suitable for publication and will be formally accepted for publication once it meets all outstanding technical requirements.

Kind regards,

Jay Saha

Academic Editor

PLOS ONE

Additional Editor Comments (optional):

Reviewers' comments:

Reviewer's Responses to Questions

**Comments to the Author**

1. If the authors have adequately addressed your comments raised in a previous round of review and you feel that this manuscript is now acceptable for publication, you may indicate that here to bypass the “Comments to the Author” section, enter your conflict of interest statement in the “Confidential to Editor” section, and submit your "Accept" recommendation.

Reviewer #1: All comments have been addressed

2. Is the manuscript technically sound, and do the data support the conclusions?

Reviewer #1: Yes

3. Has the statistical analysis been performed appropriately and rigorously? 

Reviewer #1: Yes

4. Have the authors made all data underlying the findings in their manuscript fully available?

Reviewer #1: Yes

5. Is the manuscript presented in an intelligible fashion and written in standard English?

Reviewer #1: Yes

6. Review Comments to the Author

Reviewer #1: This meta-analysis confirms that anemia in pregnancy is associated with low birth weight in selected areas of sub-Saharan Africa. The inclusion of 21 articles lends strength to this conclusion.

The authors have adequately addressed the reviewers concerns and the paper is now publishable.

7. PLOS authors have the option to publish the peer review history of their article (what does this mean? ). If published, this will include your full peer review and any attached files.

**Do you want your identity to be public for this peer review?** For information about this choice, including consent withdrawal, please see our Privacy Policy .

Reviewer #1: No

---

## [Editor Report · Acceptance letter]

PONE-D-24-38352R1

PLOS ONE

Dear Dr. Belete,

I'm pleased to inform you that your manuscript has been deemed suitable for publication in PLOS ONE. Congratulations! Your manuscript is now being handed over to our production team.

Kind regards,

on behalf of

Mr. Jay Saha

Academic Editor

PLOS ONE